# The Thermographic Analysis of the Agglomeration Process in the Roller Press of Pillow-Shaped Briquettes

**DOI:** 10.3390/ma15082870

**Published:** 2022-04-14

**Authors:** Andrzej Uhryński, Michał Bembenek

**Affiliations:** Faculty of Mechanical Engineering and Robotics, AGH University of Science and Technology, A. Mickiewicza 30, 30-059 Krakow, Poland; uhrynski@agh.edu.pl

**Keywords:** pillow-shaped briquettes, roller press, temperature distribution, thermography, briquetting pressure, briquetting

## Abstract

When the briquetting process of fine-grained material takes place in the roller press unit, the pressure reached is over a hundred megapascals. This parameter is a result, among other factors, of the geometry of a compaction unit and also the properties of the consolidated material. The pressure of the unit is not constant and the changes in value depend on a given place on the molding surface. By the process of generating different types of pressure on the surface of briquettes, their compaction is different as well. The distribution of temperature on the surface of the briquettes may determine the pressure used locally on them. Nevertheless, the distribution of stress in the briquetting material is still a subject of scientific study. However, it is known that the pressure exerted on the briquette is different for different compaction systems. The article includes authors’ further thermography studies on the classical pillow-shaped briquetting process (instead of the saddle-shaped ones that were previously conducted) of four materials (calcium hydroxide and water mixture, mill scale, charcoal fines and starch mixture, as well as a mixture of EAFD, scale, fine coke breeze, molasses, and calcium hydroxide). Immediately after the briquettes left the compaction zone, thermal images were taken of them, as well as forming rollers. Thermograms that were obtained and the variability of temperature at characteristic points of the surface of pillow-shaped briquettes were analyzed. They showed differences in temperature on the surface of briquettes. In all four cases, the highest briquette temperatures were recorded in their upper part, which proves their better densification in this part. The temperature differences between the lower and upper part of the briquettes ranged from 1.8 to 9.7 °C, depending on the mixture.

## 1. Introduction

Thermography could refer to the recording and processing of the action of light (infrared radiation), where the result is highlighted by the printing or displaying on a screen. The obtained thermogram represents the temperature distribution on the surface of the observed object. Thanks to this technique, the distribution of temperature that occurs on the surface of a given object can be determined, and (after considering proper factors and conditions at the place where the measurement is carried out) the temperature can also be measured with great accuracy. Thermography measurements belong to the group of contactless and noninvasive measurements of temperature, which, in most cases, is the basic assessment factor of their use. This method is widely used, because one can correlate many phenomena with the changes in temperature in the tested object.

The popular areas are the widely understood construction industry. It is applied there to analyze the efficiency of energy of houses to detect defects in the heating system, as well as the efficiency of introducing modernization [1]. It is also used successfully to detect the stratification of infrastructure and buildings [2]. Such a technique (due to its noninvasiveness) has also been used in the medical sciences and related scientific research with positive response [3,4,5]. It can be a quick and effective tool to detect the information necessary to diagnose rare diseases [6].

First, thermography can be used in the material removal treatment. It concerns not only the control of the temperature of the material that is being treated [7], but also the general control of the growth of temperature and its distribution in the machining zone [8], or the temperature of the surface of the chip [9]. The use of thermography to diagnose the welding process and to estimate the quality of these connections have also been mentioned [10,11]. Secondly, thermography is used in the widely understood diagnostics of machines and their components, such as CNC machine tools [12], through mechanical spreaders [13], rotating machines [14], belt conveyors [15], internal combustion engines (catalytic reactors) [16,17], bearings (temperature measurement of moving rolling elements, and one must keep in mind that baskets in bearings may be measured only with contactless methods) and bearing nodes [18,19], brakes [20], as well as electrical devices and machines [21,22,23].

There have been attempts made to correlate the tests including strength and stress, which were carried out in the material with the amount of heat that was generated [24]. Methods of measuring steel ropes including the relation between the growth of temperature and force were described [25]. The deformation of plastic and cavitation of tensile polypropylene were examined [26], and a system for the comprehensive thermomechanical analysis of materials was suggested [27]. After that, strength tests carried out during the bending test included thin-walled composite beams [28]. Then, methods were developed regarding how to use thermal imaging techniques to estimate the degree of fatigue degradation of polymeric materials (particularly epoxy composite) [29].

Apart from what we call passive thermography, there is an active one that is also used. It is a method that stimulates the subject of research, e.g., by providing a thermal impulse [30], or obtains this effect by means of acoustic activation [31]. It includes ultrasound [32] or microwaves [33]. Then, we can observe the path of its propagation using a thermal imaging camera on the surface of the object. This type of thermography is particularly important in defectoscopy [34,35].

Phenomena research by thermography is a rich source of information related to technological processes and to changes or irregularities that take place during them [36]. It allows one to control the parameters of the process and to introduce the necessary modification. For example, thermography was used, among other methods, to control the stability of the extrusion process of poly(vinyl chloride) [37]. Moreover, it has been used in textiles [38,39], tire vulcanization [40], the production of casting molds [41], the process of casting and cooling steel [42], and finally in sintering iron ores (with biomass) [43]. In one of the most recent studies concerning the wearing of the ploughshare material on the raking face when it operates in soil, the temperature distribution on the tool during this task was checked. Then, a hypothesis was formed that the amount of heat emitted in a given area on the surface of the ploughshare is related to the intensity of tribological processes [44].

Despite the extensive use of thermography, it is rarely used to study consolidation processes. The published works concern only pressure consolidation and are focused exclusively on the consolidation process in roller presses [45,46]. The main advantage of roller presses is its permanent ability to perform with a relatively low demand for energy supply [47]. The briquetting process takes place between two rollers that rotate in opposite directions that are mutual, and molding cavities are distributed in a proper way on their surfaces [48]. The agglomeration conducted in a proper way must include effective preparation of the material, its dosing [49], and proper selection of the geometry of the compaction unit [50]. Working surfaces of rollers in a traditional roller press are smooth or provided with cavities that are arranged in such a way as to be their mutual mirror images on both rollers [51,52]. In the scientific sources, the compaction unit of this type is referred to as symmetrical (Figure 1a). What prevents unfavorable phenomena from occurring during material consolidation is a mutual differentiation of the working surface of both rolls [53]. Thanks to that, an asymmetric compaction unit can be established. The manufacture of briquettes in a traditional asymmetrical compaction unit results in them being in the shape of a saddle (Figure 1b), which makes them particularly useful for materials that are difficult to briquette in a roller press [54,55].

Owing to inner friction and work during consolidation, heat is generated between the particles of the merged material that, in turn, causes a change in temperature in a certain volume of the briquette. The higher the pressures of a unit exerted on the material, the higher the consolidated material and temperature are. Therefore, the distribution of temperature on the surface of the briquettes may also be related to the pressure on the briquette that has been locally exerted [56]. The experiments that were conducted by Litstera et al. [45] included microcrystalline cellulose and, through them, they proved that the distribution of temperature in a roller compaction process (which means briquetting with flat rollers) across the width of the ribbon is not uniform. Temperature differences between the central area and the edges of the ribbon can be observed. They are linked with the inter alia with a different degree of compaction of the material (there is a higher density in the middle of the part), which results from an uneven flow of the material in the press feeder. The experiment carried out by Bembenek and Uhryński [46] involving the saddle-shaped briquette showed that the stresses and strain are greater in the upper part of the briquette, and those in the lower part are smaller, which corresponds with the briquette density [55,56].

The distribution of stress in the briquetting material has still been a subject of research. There is a lack of knowledge about the temperature distribution of briquette surfaces other than in the shape of a saddle; therefore, infrared thermography studies of the pillow-shaped briquetting process were conducted.

## 2. Materials and Methods

The thermographic research of the briquetting process in the roller press was performed using a roller press (Figure 2) with a 450 mm roller pitch diameter with an installed compaction unit for the production of pillow-shaped briquettes with a size of 40 × 30 × 20 mm and a rated capacity of 13 cm^3^ (Figure 3). There were 36 forming cavities in each of the two rows on the surfaces of the forming rollers. The outline view of the working surface and the briquette are presented in Figure 4. The roller press was equipped with a 22 kW motor with the cycloidal gear and a frequency converter that enabled infinitely variable control of the revolutionary speed of the rolls. A gravity feeder was used when all materials were consolidated. The roller revolutionary speed was 0.85 RPM, which gave a peripheral speed of the rollers equal to 0.02 m/s. It caused each briquette in the forming row to fall out approx. 2.0 s. The inter-roller gap was set to of 1 mm.

Materials of three various origins and various chemical composition and structure were selected for the tests:Material of organic origin/fuels: charcoal;Material of inorganic origin: hydrated lime;Heavy industrial waste: electric arc furnace dust (EAFD) mixtures as well as mill scale. All materials differed in bulk density, granular-metric composition, the degree of compaction necessary for consolidation, and moisture content.

Before the agglomeration process, four mixtures were prepared of the materials referred to above. They were thoroughly mixed and brought to a proper moisture, enabling them to be consolidated in a roller press, and the binders were added. The moisture content was determined by the weight method at 105 °C until a constant weight was obtained. The Vibra AJH 420 CE (Tokyo, Japan) scale was used.

### 2.1. Mixture 1 (M1)

Its composition was 83.3% calcium hydroxide manufactured by Lhoist (EN 459-1 CL 90-S) (Limelette, Belgium) and 16.7% water. The mixture was mixed in a Z-blade mixer with four rectangular mixing elements with dimensions of 190 × 90 mm and a shaft rotating speed of 55 RPM (Figure 5) for about 30 min. The moisture content of the mixture was 16.9%. The mixture was pre-compacted. The pre-compacted process involved briquetting and crushing the consolidated briquettes to a size below 10 mm.

### 2.2. Mixture 2 (M2)

The mixture consisted of mill scale, with a grain size up to 5 mm with 4% water addition. Its moisture content was 4.2%.

### 2.3. Mixture 3 (M3)

The pre-compacted charcoal fines were mixed with 4% of starch. The moisture content of the mixture was 26.9%. The pre-compacted process involved briquetting and crushing the consolidated briquettes to a size below 10 mm.

### 2.4. Mixture 4 (M4)

The mixture contained 47.7% of EAFD, 36.7% of scale, 7.3% fine coke breeze, 5.5% 80° Bx molasses, and 2.8% calcium hydroxide. The last two ingredients acted as a binder. The mixture was mixed for approx. 10 min in the double-arm Z-blade mixer. Its moisture content was 4.6%.

The FLIR T335 Thermal Imaging Camera (Wilsonville, OR, USA) was used for the tests. The operating temperature range of the camera was: −20 to +650 °C, and the temperature measurement error was: ±2 °C or ±2% of the measurement value. The camera was equipped with a microbolometric matrix with a resolution of 320 × 240 pixels, a 25° × 18.75° lens, and a 0.05 °C Noise Equivalent Temperature Difference (NETD) sensitivity. Before the tests, it was necessary to calibrate the camera. The ambient temperature during tests was 18.5 °C. Access to daylight and artificial light to the measuring station was eliminated in the laboratory. In addition, a special shield was made (Figure 6) to block the inflow of the light to the working area of the thermal imaging camera, which eliminated the reflection of radiation disturbing the test result and ensuring repeatability of the distance between the camera and the tested briquette. Each time the briquette was produced in a roller press, it was caught and transferred to a plywood plate, using heat-insulating gloves, immediately after leaving the compaction unit. Then, the briquettes and the pad were placed in a measuring station and thermal images were taken. The time between catching the briquettes and taking the photo was about 3 s. After each test, the plywood pad was changed to make the test conditions reproducible as the pads got hotter. The pads got hotter from the briquettes as time went on. It was possible to transfer the briquette to the plywood pad directly after it was removed from the compaction unit, while maintaining a precise control of briquette top-bottom, front-back orientation due to the low peripheral speed of the rollers (0.02 m/s). The tests of briquetting with the higher peripheral speed were unsuccessful due to the lack of possibility of controlled catching of the briquettes in the right orientation [46]. The images of the briquettes were taken in such a way that their “top” (Figure 7) was always located up the top edge of the image. Due to the fact that both surfaces of the briquettes were symmetrical, images were taken only on the front side of the briquettes, and not as it was in previous tests (the front and back surfaces are different in the case of saddle-shaped briquettes).

From each mixture, up to 10 thermal images of the briquettes were taken, out of which 3 with the highest maximum temperature were selected for further analysis (Figure 7b). The FLIR Quick Report program (Wilsonville, OR, USA) was used to analyze the images. First, the maximum and minimum temperatures were determined for each type of briquette, and its measurements were averaged.

To examine the temperature distribution on the briquette surface, a special grid shown in Figure 8 was fitted into the thermograms. The measuring points were spaced 6.5 mm from each other. For each image in the vertical axis of the briquette at 7 points, temperature values were read.

## 3. Results and Discussion

The first stage of the research was to produce a pilot batch of briquettes from each of the mixtures. From all the materials used in the tests, good-quality briquettes that did not crumble were obtained. The briquettes were then seasoned at ambient temperature for 24 h. Then, the emissivity of the integrated materials was determined. The briquettes were put into a previously heated SPT-200 furnace (ZUT Colector, Krakow, Poland) in order to obtain a temperature higher than the ambient temperature. After removing the briquettes from the furnace, their surface temperature was measured using a Kyoritsu KEW 1011 multimeter and K-type 8216 thermocouple (Kyoritsu, Tokyo, Japan); it was 49 °C. Using the FLIR QuickReport v. 1.2 SP1 program, the emissivity—the ability to emit thermal radiation—of individual briquettes was determined according to the following algorithm:Loading a thermogram in the QuickReport v. 1.2 SP1 program.Selecting the briquette for analysis.Readout of the maximum temperature on the surface of the selected briquette (Figure 9).Selecting the emissivity so that the maximum temperature on the surface of the selected briquette on the thermogram was equal 49 °C.Reading the emissivity of a given briquette.Repeating the operation for other briquettes.

The emissivity results of individual materials are presented in Table 1.

The range of the obtained emissivity values of briquettes obtained from the material mixtures for the temperature of 49 °C (±0.1 °C) was 0.69–0.91. The briquette made of calcium hydroxide had the highest emissivity (ε = 0.91), while the charcoal briquette had the lowest one (ε = 0.69). The values of the reflection coefficients of the briquettes were not too high, which allowed the measurement of the temperature of their surface without burdening the measurement with large errors caused usually by reflection coefficients that are too high, i.e., reflected heat radiation. The above cases of emissivity were taken into account appropriately for each type of briquette.

Then, the actual tests began. The results of the minimum and maximum temperature for each type of briquette on the front sides and its average value are presented in Table 2 and in Figure 10.

In Figure 10, the summary graph of the minimum surface temperatures, it can be seen that the lowest temperature on the surface was obtained for the briquettes made of Mixture 3 (pre-compacted charcoal fines mixed with 4% of starch). The minimum temperature of these briquettes indicated that, in this case, a forming process was taking place at a given location on the surface rather than briquetting. In Figure 10, the summary graph of the maximum surface temperatures, it can be seen that the highest temperature on the surface was obtained for the briquettes made of Mixture 2 (mill scale). The maximum temperature of these briquettes was 13.8 °C higher than the ambient temperature. The results of the temperature distribution on the briquette surfaces for each type of briquette are presented in Table 3 and in Figure 11.

From a practical point of view, the best solution would be a situation in which the curves would not be obtained, but lines perpendicular to the temperature axis (X axis), which would mean that the pressure in the entire briquette was equal and the briquettes would have the same physical properties throughout the volume. However, analyzing the obtained results and the curves of temperature variation on the front surface of the briquettes at characteristic points, it can be concluded that the curves of temperature variation for each of the materials from which the briquettes were obtained were similar to each other. Similar to saddle-shaped briquettes [46], the maximum temperature values were obtained in all cases in point 2, except for the M3 mixture, where the temperature reached the maximum value in point 1. This leads to the conclusion that the area of maximum temperature and thus the area of the highest unit pressure was in the upper part of the briquette. This was probably due to the inability of the material to escape from the forming cavities during the roll rotation process and closing the formed volume of material in the forming cavities. The minimum temperature values for each of the briquettes were read for each time in point 7. This led to the conclusion that the place where the pressure took the smallest value was the lower part of the briquette. All curves of temperature variation, except the curve for the M3 mixture, were characterized by an increase in temperature from point 1 to point 2, followed by a linear decrease in temperature to point 7. The lowest temperature values at all points were recorded for briquettes obtained from the M3 mixture—pre-compacted charcoal with the addition of 4% potato starch. The highest temperature values were recorded at all points for the briquette obtained from the M2 mixture of metallurgical scale with a grain size up to 5 mm and with the addition of 4% water. The temperature distribution on the surface of pillow-shaped briquettes correlates with research conducted by other scientists. Hryniewicz proved that the greatest unit pressure was exerted on the pillow-shaped briquette in its upper part [57], while Loginow et al. showed on an experimental stand that the highest deformations of the material occurred in the upper part of this type of briquette [52].

Analyzing the thermograms of briquettes, it can be seen that the highest temperature values were present in one of the upper corners of the briquette, e.g., in Figure 12, in the upper left corner of the briquettes. This proved that the temperature distribution on the briquette surface also depended on the arrangement of other forming cavities on the working surface of the rollers. These briquettes were taken from one and the same row of cavities of the forming rings. In turn, the thermograms presented in Figure 13 show two briquettes made of the same material (M2 mixture) but taken from two different rows of cavities of the forming rollers.

It has been found out that this temperature distribution correlates with the wear of the forming ring with multiple rows [58]. It can be seen that the temperature in the upper part of the briquettes in the outermost rows of the forming cavities, especially on the outer side, was lower and the wear of the rings was lower in these places. This was due to the possibility of the compacted material escaping from the compaction zone in these places. In other rows of forming cavities, the escaping material became interlocked with the material escaping from other cavities, preventing it from escaping from the compaction zone and thereby increasing the unit pressure. A lower density of briquettes in these places may also result in the slowest flow of material occurring at the walls of the feeder side sealings. A comparison of the industrial wear of the molding ring and the obtained thermogram is shown in Figure 14. The rollers worked for about 1100 h in an industrial environment. The material subject to briquetting was copper ore concentrate with sulfite lye with a moisture of about 4.0–4.5%. Theoretically, it is possible to reduce the wear of so-called dead spots located on the outermost rows of the molding cavities, by making special cavities in these places. However, according to authors, such a solution does not make sense, as it will lead to increase the amount of undersized material to be returned to the briquetting process.

## 4. Conclusions

As in the case of previous research on saddle-shaped briquettes [46], it has been proven that thermography can also be used to analyze the processes of briquetting of fine-grained materials where the pillow shape of briquettes is obtained. The temperature in the upper part of the briquette for each type of the consolidated material was highest. The temperature distributions coincided with the bulk material deformation in the forming cavity during the briquetting process in symmetrical compaction unit. The higher local briquette temperatures in a given place corresponded to a better density in that place. Therefore, it can be concluded that in the case of the pillow-shaped briquette, the stresses and strain were greater in the upper part of the briquette, and those were smaller in the lower part. The presented work also corresponds with the results obtained by Hryniewicz who proved that the greatest unit pressure is exerted on the pillow-shaped briquette in its upper part. A very important aspect of the work is the demonstration, by thermographic studies, of uneven wear of the extreme cavities forming.

To sum up, measuring the temperature by means of thermovision on the briquette surface after the briquetting process can be an indirect method for determining, for example, the degree of briquette compaction or the pressure generated in the forming cavity during the consolidation process. It can also be used to select the appropriate geometry of the forming cavities, enabling one to obtain properly compacted briquettes.

The authors predict the continuation of the research in order to determine the exact relation between the applied pressure and the local temperature of the briquette; therefore, a special stand is currently being built for this purpose.

## Figures and Tables

**Figure 1 materials-15-02870-f001:**
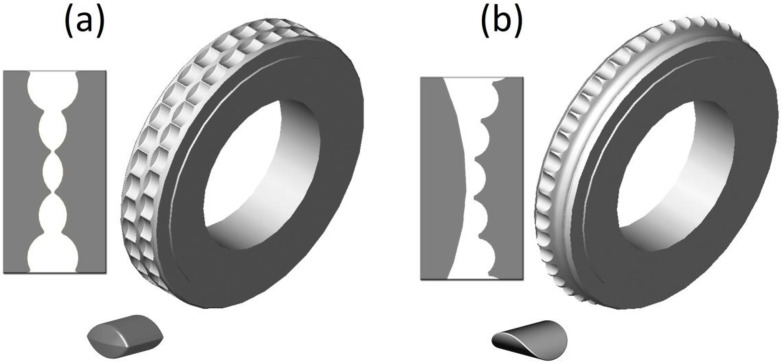
The view of the roller press forming rollers with a working surface to produce briquettes (**a**) in the shape of a pillow and (**b**) in the shape of a saddle.

**Figure 2 materials-15-02870-f002:**
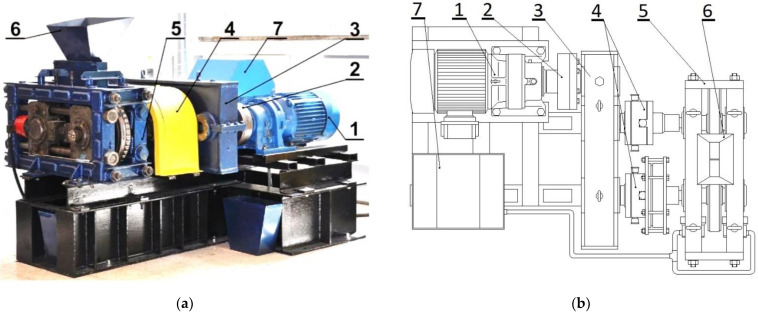
The laboratory roller press LPW 450: (**a**) the view [47], (**b**) the scheme: (**1**) gear motor with a cycloidal transmission; (**2**) flexible clutch; (**3**) gearbox; (**4**) enclosure of Oldham couplings and friction clutch; (**5**) molding rollers cage; (**6**) gravity feeder; (**7**) hydraulic system of sliding roller support.

**Figure 3 materials-15-02870-f003:**
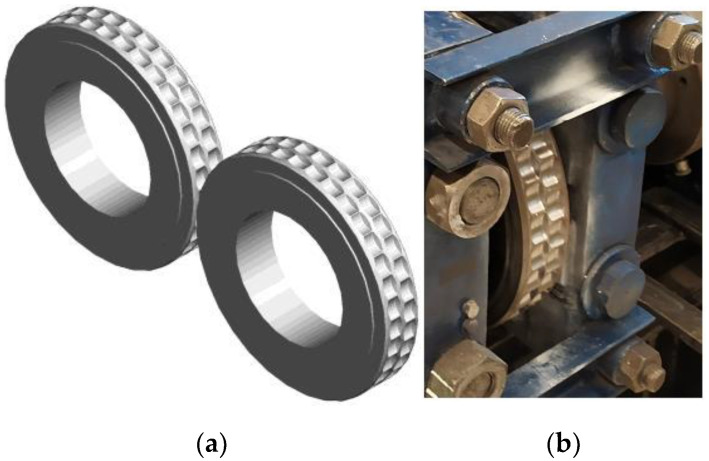
The rollers used in roller press compaction unit with a pillow-shaped briquette: (**a**) scheme of compaction unit and (**b**) the roller installed in the roller press.

**Figure 4 materials-15-02870-f004:**
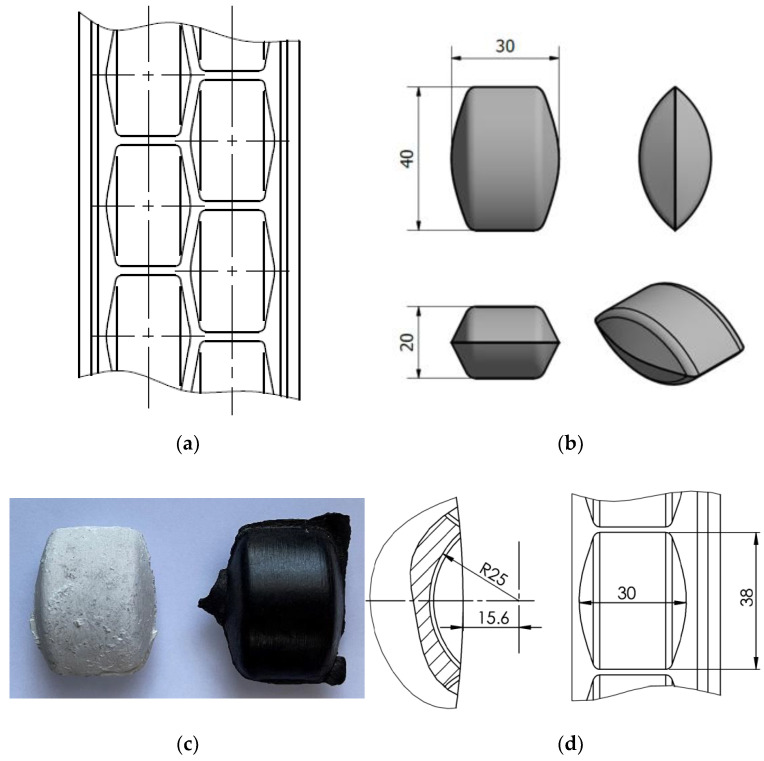
Geometry of molding cavities on the working surface of rollers used for tests: (**a**) front view or working surface, (**b**) the dimension of the consolidated briquette, (**c**) the real briquettes, and (**d**) dimensions of the forming cavity. (unit: mm).

**Figure 5 materials-15-02870-f005:**
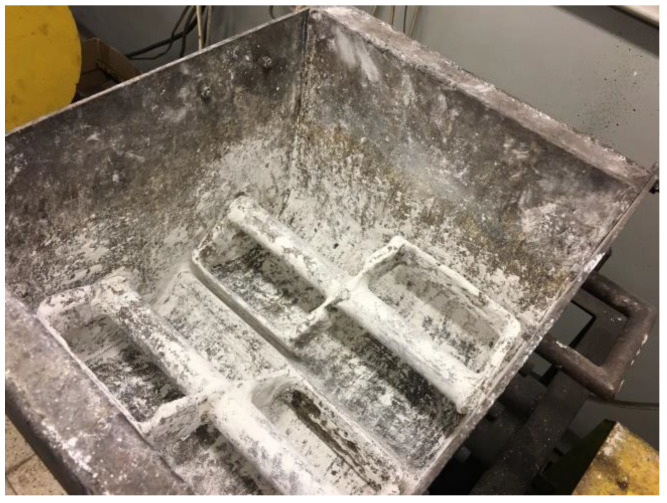
The view of the Z-blade mixer used in the test for material preparation [46].

**Figure 6 materials-15-02870-f006:**
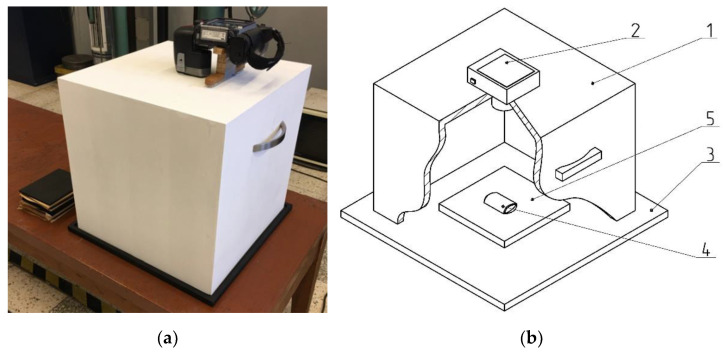
A specially constructed station for taking images of briquettes, which eliminates the effects of external radiation: (**a**) the image [46], (**b**) the scheme: (**1**) cover; (**2**) thermal imaging camera; (**3**) base; (**4**) briquette; (**5**) plywood pad.

**Figure 7 materials-15-02870-f007:**
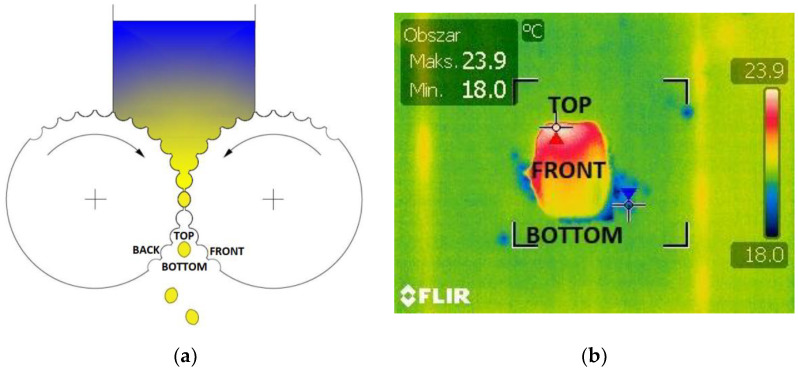
The briquettes arrangement: (**a**) in the compacting unit of the roller press; (**b**) on the plywood pad.

**Figure 8 materials-15-02870-f008:**
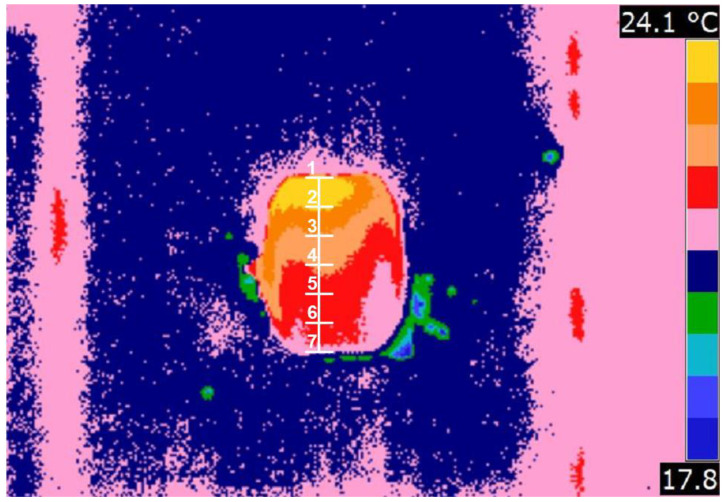
Thermogram processed in FLIR Quick Report with measuring points where temperature readings were taken to prepare the temperature distribution on the briquette surfaces: Point 1—end of briquette consolidation; Point 7—start of briquette consolidation.

**Figure 9 materials-15-02870-f009:**
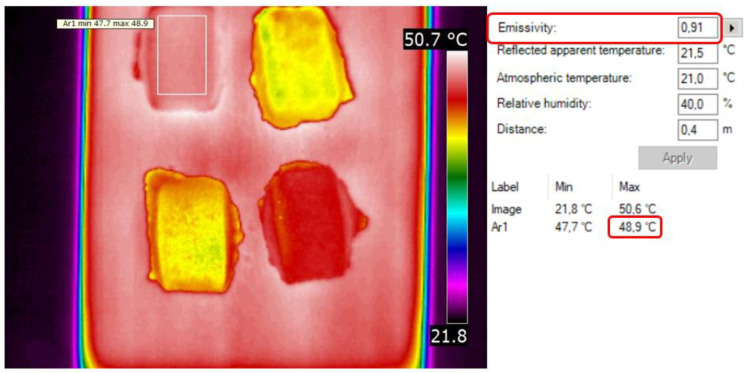
Exemplary determination of briquette emissivity from M1 mixture in FLIR QuickReport 1.2.

**Figure 10 materials-15-02870-f010:**
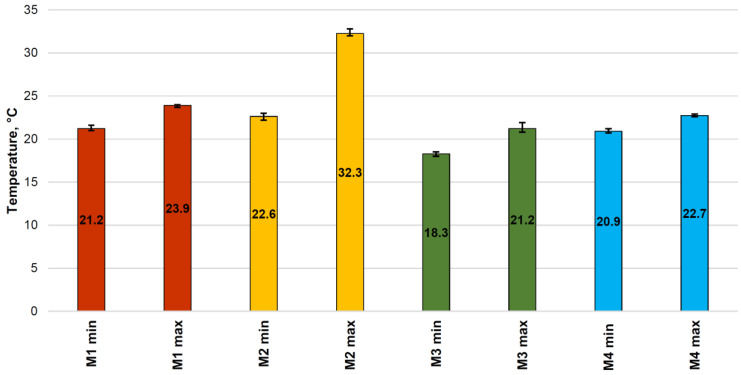
The graphs of average minimum and maximum temperature measurements on the surface of pillow-shaped briquettes after briquetting.

**Figure 11 materials-15-02870-f011:**
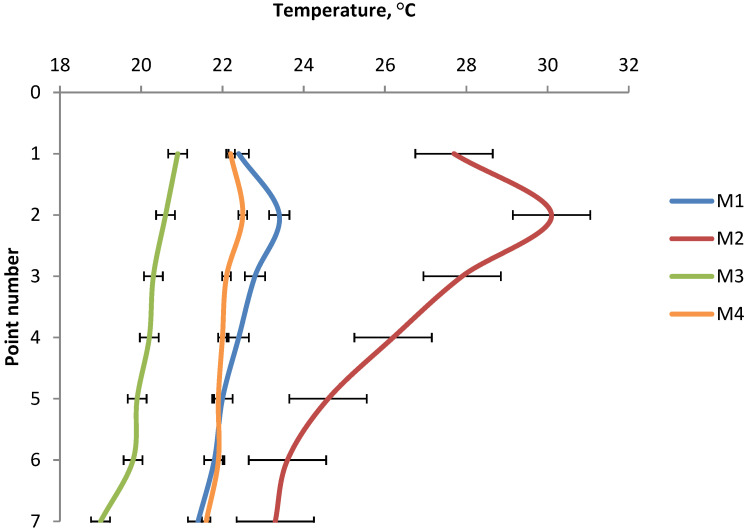
The temperature distribution on the front side of briquette surfaces.

**Figure 12 materials-15-02870-f012:**
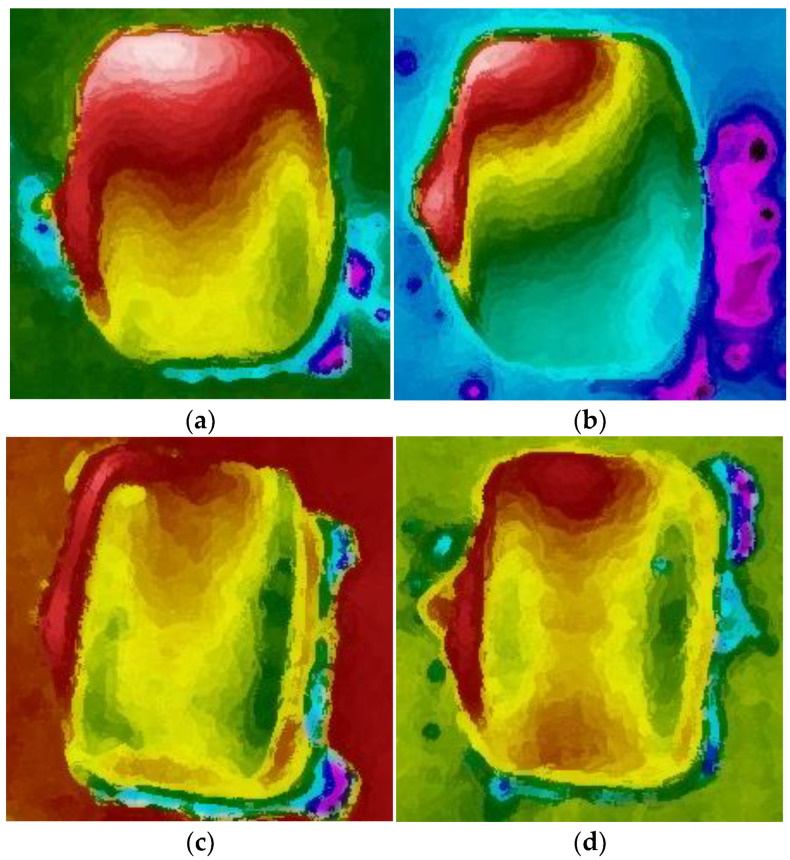
Briquette thermograms showing the temperature gradient across its surface: (**a**) M1, (**b**) M2, (**c**) M3, and (**d**) M4.

**Figure 13 materials-15-02870-f013:**
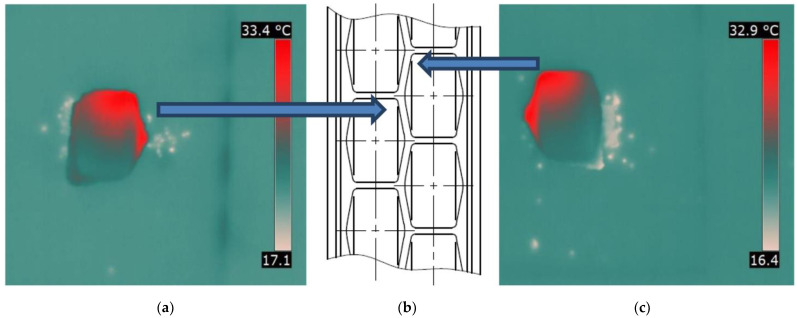
Comparison of thermograms of briquettes obtained from the M2 mixture (mill scale) taken from two different rows of forming cavities: (**a**) left row briquette, (**b**) scheme of working surface, and (**c**) right row briquette.

**Figure 14 materials-15-02870-f014:**
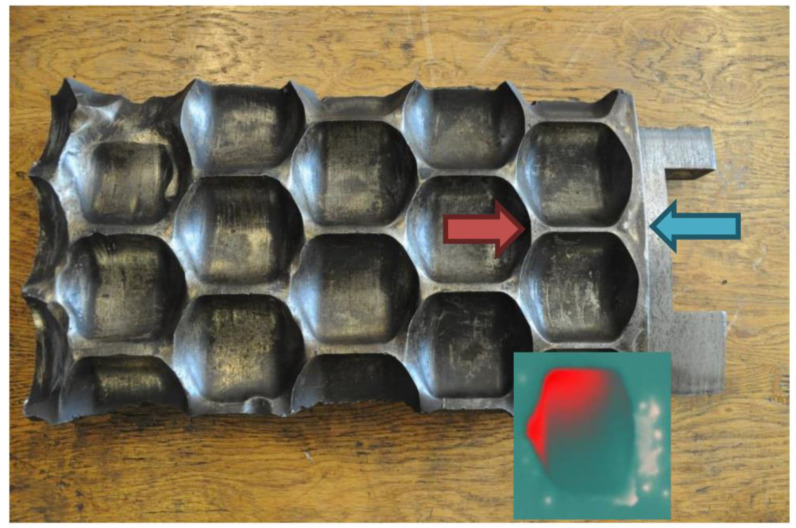
Fragment of a worn-out industrial molding ring compared to thermogram of the briquette.

**Table 1 materials-15-02870-t001:** Emissivity of briquettes from individual materials at 49 °C (±0.1 °C).

	Temp. Maximum for Emissivity ε = 0.95, °C	Emissivity, -	Temp. Maximum, °C	Reflectance Coef. (1 − ε), -
M1 (calcium hydroxide)	47.9	0.91	48.9	0.09
M2 (mill scale)	42.6	0.70	49.1	0.30
M3 (charcoal)	42.3	0.69	49.0	0.31
M4 (EAFD mixture)	45.5	0.82	48.9	0.18

**Table 2 materials-15-02870-t002:** The results of minimum and maximum temperature measurements on the entire surface of briquettes after briquetting.

Side	M1, °C		M2, °C		M3, °C		M4, °C	
Minimum	21.6	21.2	22.2	22.6	18.0	18.3	20.7	20.9
	21.1		22.7		18.3		21.2	
	21.0		23.0		18.5		20.8	
Maximim	23.7	23.9	32.0	32.3	20.8	21.2	22.6	22.7
	24.0		32.8		21.0		22.7	
	24.0		32.0		21.9		22.9	

**Table 3 materials-15-02870-t003:** The temperature distribution on the front side of briquette surfaces in the characteristic points.

Point	M1, °C		M2, °C		M3, °C		M4, °C	
	22.3		27.7		21.3		22.3	
1	22.9	22.4	27.5	27.7	20.9	20.9	22.5	22.2
	22.0		27.9		20.5		21.9	
	23.5		29.3		20.9		22.4	
2	23.3	23.4	31.5	30.1	20.3	20.6	22.4	22.5
	23.5		29.6		20.7		22.6	
	22.8		27.1		20.4		22.1	
3	22.9	22.8	28.9	27.9	20.2	20.3	22.0	22.1
	22.8		27.6		20.3		22.1	
	22.3		25.3		20.1		22.0	
4	22.5	22.4	27.0	26.2	20.1	20.2	21.9	22.0
	22.4		26.3		20.3		22.0	
	21.9		24.1		19.9		21.8	
5	22.1	22.0	25.1	24.6	19.8	19.9	21.8	21.9
	21.9		24.7		20.1		22.0	
	21.7		23.2		19.9		21.8	
6	21.8	21.8	23.9	23.6	19.5	19.8	21.9	21.9
	21.8		23.8		20.0		22.1	
	21.3		23.0		19.0		21.7	
7	21.5	21.4	23.3	23.3	18.0	19.0	21.3	21.6
	21.3		23.5		19.9		21.8	

## Data Availability

The data presented in this study are available upon request from the corresponding author.

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
