# Peer review of "The Thermographic Analysis of the Agglomeration Process in the Roller Press of Pillow-Shaped Briquettes"

_materials, 2022, doi:10.3390/ma15082870_

Round 1

Reviewer 1 Report

This manuscript presented a thermographic analysis of the Agglomeration Process in the Roller Press of Pillow-Shaped Briquettes. With the application of thermography, the distribution of pressure during the processing of briquette was revealed. The findings is in agreement with what happened in high pressure grinding roller (HPGR) in ore comminution field. It was found that the pressure at the middle of roller is higher than that at the edge in HPGR studies. This manuscript improved the understanding of the pressure distribution during the processing of briquettes and may helpful to improve the production of briquettes.

I recommend to publish with minor revision.

Some minor problems includes:

Line 125: what is “IR” thermography?

Section 3 paragraph 1: I guess the aim to measure emissivity is for calibration, but the authors did not confirm it is for calibration.

Table 2: what were the locations of minimum and maximum temperature measurement?

Table 2: not maksimim, but maximum

Figure 9 and 10: a box plot would be better to replace these two figures.

Table 3: what is the result of all 10 measure samples rather than the hottest 3 samples?

Figure 13: sub-picture c’s name should be “right” row briquette

Author Response

Thank you very much for taking the time to read our manuscript thoroughly and make recommendations for its correction and improvement.

Remark 1

Line 125: what is “IR” thermography?

Response:

IR means infrared. This has been corrected in the article.

Remark 2

Section 3 paragraph 1: I guess the aim to measure emissivity is for calibration, but the authors did not confirm it is for calibration.

Response:

The emissivity of briquettes was determined so that  to obtain accurate results in terms of temperature values, as they were the most important parameter when performing non-contact tests. In order to make as few error as possible when determining this parameter (between different types of mixtures), it was necessary to test all types of mixtures at the same time, which required their previous merging. During the process of merging them, the main tests were carried out, and the briquettes obtained as a result of merging them were subjected to the emissivity test. The emissivity values determined in this way were introduced during the processing of thermograms in the FLIR QuickReport 1.2 program, correcting them appropriately for each tested material and in this way  obtaining the correct temperature results.  Either by introducing the emissivity value before testing or by correcting this parameter during the subsequent processing of thermograms (if it was incorrectly given) one can obtain  the same correct results.

Remark 3

Table 2: what were the locations of minimum and maximum temperature measurement?

Response:

The minimum and maximum temperatures were determined from the entire surface of the briquette.

Remark 4

Table 2: not maksimim, but maximum

Response:

This has been corrected in the article.

Remark 5

Figure 9 and 10: a box plot would be better to replace these two figures.

Response:

This has been corrected in the article.

Remark 6

Table 3: what is the result of all 10 measure samples rather than the hottest 3 samples?

Response:

This is the methodology we adopted in the previous article and we wanted to keep it so that we could later compare the results of the research with each other.

Remark 7

Figure 13: sub-picture c’s name should be “right” row briquette

Response:

This has been corrected in the article.

Reviewer 2 Report

I checked the manuscript 1637149. My opinions are as follows,

  1. Line 105 of text; The number of Figure 2 is mistake.  Please correct to Figure 1.
  2. Line 140, Figure 2; Please indicate the position or layout of Thermal Imaging Camera, because the camera is most important system for this paper.
  3. Line 220, Figure 6; Besides this photo, the layout of the thermal camera, the sample briquette and the plywood plate without the special shield. Readers cannot image of inside of the shield.
  4. Line 260, Figure 9; Please explain more detail. Can I understand that there are four briquettes of different temperature and mostly uniformly distribution (no temperature gradient) for M1.
  5. Table 2; “Maksimim” have to be correct to “Maximum”.
  6. Line 279; There are two Figure 9. Please correct the numbers of figure and text after this.
  7. Line 301, Figure 11; Please indicate which of the four temperature distribution is ideal for production of the briquette and that reason, if it possible.
  8. Line 327, Please show the example of cavity on the roller by photo which indicate the size, depth and surface appearance.
  9. Line 355-356, Authors point outed the case of “greater-degradation of the forming rollers”. Please show how to improve the design the production system in order to prevent the degradation, if it possible.
  10. Line 361-375, Conclusions; Authors use reference [46],[52],[57] and [58] in conclusions, but these parts should be moved to the discussion of the main text. I think the conclusion should be consisted of only authors result, data and ideas.   

The above are all my opinions.  17/March/2022

Author Response

Dear Reviewer,

Thank you very much for taking the time to read our manuscript thoroughly and making  recommendations for its correction and improvement. We have read the comments carefully and have responded to all of them.

Remark 1

  • Line 105 of text; The number of Figure 2 is mistake.  Please correct to Figure 1.

Response:

This has been corrected in the article.

Remark 2

  • Line 140, Figure 2; Please indicate the position or layout of Thermal Imaging Camera, because the camera is most important system for this paper.

Response:

In our researches, the positioning of the camera relative to the roller press did not have any effect on the test results. The thermal imaging camera was positioned independently of the roller press and installed in the special station (shield) shown in Figure 6. The briquettes were transported to the station using gloves.

Remark 3

  • Line 220, Figure 6; Besides this photo, the layout of the thermal camera, the sample briquette and the plywood plate without the special shield. Readers cannot image of inside of the shield.

Response:

Thank you for this remark. In order to illustrate the measurements with the special shield better, we have included an additional diagram in the article (Fig. 6b).

Remark 4

  • Line 260, Figure 9; Please explain more detail. Can I understand that there are four briquettes of different temperature and mostly uniformly distribution (no temperature gradient) for M1.

Response:

This photo shows an additional test after the production of the briquettes to determine their emissivity. In this case, all types of briquettes were heated in the furnace to a temperature of 49 degrees. The temperature gradient occurred only in the case of freshly produced briquettes.

Remark 5

  • Table 2; “Maksimim” have to be correct to “Maximum”.

Response:

This has been corrected in the article

Remark 6

  • Line 279; There are two Figure 9. Please correct the numbers of figure and text after this.

Response:

This has been corrected in the article.

Remark 7

  • Line 301, Figure 11; Please indicate which of the four temperature distribution is ideal for production of the briquette and that reason, if it possible.

Response:

We added the following  information to the article: From a practical point of view, the best solution would be a situation in which the curves would not be obtained, but lines perpendicular to the temperature axis (X axis), which would mean that the pressure in the entire briquette was equal and the briquettes would have the same physical properties throughout the volume .

Remark 8

  • Line 327, Please show the example of cavity on the roller by photo which indicate the size, depth and surface appearance.

Response:

A drawing of the dimensions of the forming cavity has been added to the article (Fig.4c)

Remark 9

  • Line 355-356, Authors point outed the case of “greater-degradation of the forming rollers”. Please show how to improve the design the production system in order to prevent the degradation, if it possible.

Response:

Theoretically, it is possible to reduce the wear so-called dead spots located on the outermost rows of the moulding cavities, by making special cavities in these places. However, according to authors, such a solution does not make sense, as it will lead to an increase of the amount of undersized material to be returned to the briquetting process.

Remark 10

  • Line 361-375, Conclusions; Authors use reference [46],[52],[57] and [58] in conclusions, but these parts should be moved to the discussion of the main text. I think the conclusion should be consisted of only authors result, data and ideas.

Response:

This has been corrected in the article.

Reviewer 3 Report

This paper focuses on the thermographic analysis during the production of briquettes which provides an insight into process pressure maps and stress distribution. Few small items to be addressed before publication. The manuscript can be improved by considering the following: 

  • Please update the abstract with the outcomes and quantitative results. For example, please compare and quantify the temperature difference on the surface of briquettes.
  • In the abstract, the authors mentioned about four cases. It is not clear whether four cases are related to the mixture of 4 materials. Please clarify.
  • Please consolidate the section about thermography and its various applications. These references are not mostly related to the work presented in this paper. Authors may consider the section focuses on the rolling process itself.
  • Please check the sentence construction throughout the manuscript, e.g.
    • Despite the extensive use of thermography, “it” is rarely used to study consolidation processes (it to be added)
  • In the introduction, please mentioned the gaps in the literature and what are the objectives of this study.
  • A schematic representation of Figure 2 would be more beneficial for understanding the process and arrangement.
  • Please explain the basis for the selection of four mixtures. The objectives for preparing these mixtures are also not clear.
  • Please use the full form of NETD and the type of thermocouple used for this measurement.
  • Please check whether - 49°C or 49°C.
  • Please include the error bar in Fig 11 as well.
  • Please also highlight the developed method’s industrial applications/how industries can be benefitted from it.

Author Response

Dear Reviewer,

Thank you very much for taking the time to read our manuscript thoroughly and make recommendations for its correction and improvement. We have read the comments carefully and have responded to all your comments.

Remark 1

Please update the abstract with the outcomes and quantitative results. For example, please compare and quantify the temperature difference on the surface of briquettes.

Response:

Thank you for the remark. We have included this in the abstract.

Remark 2

In the abstract, the authors mentioned about four cases. It is not clear whether four cases are related to the mixture of 4 materials. Please clarify.

Response:

Thank you for the remark. We have made changes in the abstract to describe the mixtures use better.

Remark 3

Please consolidate the section about thermography and its various applications. These references are not mostly related to the work presented in this paper. Authors may consider the section focuses on the rolling process itself.

Response:

A large number of references resulted from the fact that we wanted to show that despite the fact that thermography is widely used, it is not used in the  materials briquetting which makes our tests innovative. In fact, there are only two publications available on the subject of the thermography of the briquetting process in roll presses, and this is probably not enough to make a meaningful introduction out of it.

Remark 4

  •  
  • Please check the sentence construction throughout the manuscript, e.g.
    • Despite the extensive use of thermography, “it” is rarely used to study consolidation processes (it to be added)
    •  

In the introduction, please mentioned the gaps in the literature and what are the objectives of this study.

Response:

This has been corrected in the article.

Remark 5

A schematic representation of Figure 2 would be more beneficial for understanding the process and arrangement.

Response:

An additional scheme in the article was added (Fig. 2b).

Remark 6

Please explain the basis for the selection of four mixtures. The objectives for preparing these mixtures are also not clear.

Response:

All materials differed in bulk density, granular-metric composition, the degree of compaction necessary for consolidation and moisture content. – we added it to the article.

The choice of materials for briquetting was also dictated by the fact that the mixtures are given to briquetting for the following purposes:

  • charcoal - making briquettes for the barbecue
  • hydrated lime for the preparation of sorbent
  • scale - utilization by briquetting and returning the briquette to the oxygen converter
  • EAFD mixtures - sintering briquettes in a shaft furnace to separate zinc from iron

Remark 7

Please use the full form of NETD and the type of thermocouple used for this measurement.

Response:

This has been corrected in the article.

Remark 8

Please check whether - 49°C or 49°C.

Response:

This has been corrected in the article.

Remark 9

Please include the error bar in Fig 11 as well.

Response:

The error bar has been added to the Figure 11.

Remark 10

Please also highlight the developed method’s industrial applications/how industries can be benefitted from it.

Response:

Due to its different properties, each material requires an appropriate size and volume of the briquette for consolidation. Method can be used to select the appropriate geometry of the forming cavities, enabling one to obtain properly compacted briquettes.

Reviewer 4 Report

Dear Authors,

The paper covers an important issue of the process of briquette preparation. The results show the thermography analysis as a very useful tool, to characterize the briquette's defects and anticipate possible problems in the downstream processes.
The introduction is well written and explains the thermography technique in detail. Is possible to improve the methodology including the particle size distribution of the different mixtures M1 to M4, and some pictures of the briquettes.
Some typing errors are highlighted in the pdf document as the word "maksimim".

Regards

Author Response

Dear Reviewer,

Thank you very much for taking the time to read our manuscript thoroughly and make recommendations

The pictures of the briquettes were added as well as typos were corrected.

Reviewer 5 Report

  1. The topic of the article is current and corresponds to the profile of the journal.
  2. The authors used some modern research software and equipment.
  3. The authors used in the article photographs (Figs. 2, 5, 6) identical to those in Bembenek, M .; Uhryński, A. Analysis of the temperature distribution on the surface of saddle-shaped briquettes consolidated in the roller press. Materials 2021, 14, 1770, but without mentioning that these photographs are taken from a work published in 2020. In the 2021 paper, for the photograph in Figure 2, there is a reference to a reference (a work previously published by the first author, but which, surprisingly, does not contain the image in figure 2 of the paper published in 2021!), while in the paper proposed for publication in 2022 no references appears. It would be necessary either to make other images or to include references to the work from 2021, affecting the level of originality of the work proposed in 2022.
  4. The definition of thermography ("Thermography is a concept including the process of recording and processing light in the infrared spectrum") does not seem the most appropriate. Thermography could refer to the recording and processing of the action of light (infrared radiation), the result is highlighted by the printing or displaying on a screen.
  5. In Figure 6, the authors highlight the appearance of a "special shield" (the word ”special” is misspelled "spcial"), but there is relatively little information on the contents of this "special shield". In the legend of Figure 6, the name "a specially constructed station for taking images during measurements" was used. It would be useful to have at least a schematic representation of a cross-section by "special shiel" (or "specially constructed station"?), to see to what extent the equipment fulfills the functions mentioned by the authors. Secondly, I think that the determination of the temperature values ​​was done by analyzing the thermographic images, so in a space outside the "special shield". Why then does the legend of Figure 6 mentions "taking images during measurements"?
  6. In Figure 8, distinct temperature values ​​appear to be highlighted on the briquette. Can an arrow be inserted to highlight the direction along which the briquette is advancing, in order to correlate the temperature values ​​with the initiation and development of the briquetting process, respectively?
  7. Figures were misspelled, the first Figure 2 being, in fact, Figure 1.
  8. In lines 319-323, the authors state that “the temperature distribution on the surface of pillow-shaped briquettes correlates with research conducted by other scientists. Hryniewicz proved that the greatest unit pressure is exerted on the pillow-shaped briquette in its upper part [57], while Loginow et al. showed on an experimental stand that the highest deformations of the material occur in the upper part of this type of briquette [52]”. In fact, the results obtained by the authors of the proposed article seem to be somewhat consistent only with the results obtained by Hryniewicz.
  9. In Chapter 3. Results and Discussion, the discussion is mainly concerned with revealing the observed aspects, and only to small extent explanations or hypotheses are offered regarding the observed aspects.
  10. The formulation “The popular areas which successfully uses thermography is widely understood construction industry.” is confusing (The popular areas are widely ...).

Other less well-expressed forms in English: " due to it being non-invasive", ”in which case they range from CNC machine tools [12], through mechanical spreaders [13], rotating machines [14], belt conveyors [15], internal combustion engines (catalytic reactors) [16,17], not to mention bearings (temperature measurement of moving rolling elements, and one must keep in mind that baskets in bearings may be measured only with contactless methods”, ”Methods of measuring steel ropes were described which included the relationship between the growth of temperature and force”, “Phenomena concerning thermography a rich source of information related to technological processes and to changes or irregularities which take place during them”, etc.

  1. Authors need to pay more attention to editing the article and expressing it in English.

Thus, in lines 60 and 6,2 there are two formulations that start, both, with “After that, ...”.

It can be written "extrusion process of poly(vinyl chloride) " instead of "extrusion process of poly (vinyl chloride)", "it is rarely used to study consolidation processes." instead of “is rarely used to study consolidation processes.”, “a special” instead of “a spcial”, in line 193, “in Table” instead of “in Tables” in line 295, etc.

Line 254 reads "was equal -49 ° C." I think the expression is wrong, the temperature value being + 49 ° C.

In lines 270, 325, after the abbreviation "i.e." a comma must also be placed.

In the case of some of the figures (Figs. 2, 10, 14) no dot was placed at the end of the figures' legends.

There is a convention that some punctuation marks should be placed immediately after a word, without a blank space appearing. In lines 121, 122, etc., a blank space was placed before the comma.

The way in which the titles of some articles are included in the list of bibliographic references is uneven (in some cases, the so-called important words were written with the first letter as a capital letter - for example, references nos. 2, 42, while in other cases, except for the first letter of the title, only lowercase letters were used. In the list of bibliographic references, abbreviations of journal titles were used only in a few cases.

Author Response

Dear Reviewer,

Thank you very much for taking the time to read our manuscript thoroughly and make recommendations for its correction and improvement.

Remark 1

The authors used in the article photographs (Figs. 2, 5, 6) identical to those in Bembenek, M .; Uhryński, A. Analysis of the temperature distribution on the surface of saddle-shaped briquettes consolidated in the roller press. Materials 2021, 14, 1770, but without mentioning that these photographs are taken from a work published in 2020. In the 2021 paper, for the photograph in Figure 2, there is a reference to a reference (a work previously published by the first author, but which, surprisingly, does not contain the image in figure 2 of the paper published in 2021!), while in the paper proposed for publication in 2022 no references appears. It would be necessary either to make other images or to include references to the work from 2021, affecting the level of originality of the work proposed in 2022.

Response:

This has been corrected in the article.

Remark 2

The definition of thermography ("Thermography is a concept including the process of recording and processing light in the infrared spectrum") does not seem the most appropriate. Thermography could refer to the recording and processing of the action of light (infrared radiation), the result is highlighted by the printing or displaying on a screen.

Response:

This has been corrected in the article.

Remark 3

In Figure 6, the authors highlight the appearance of a "special shield" (the word ”special” is misspelled "spcial"), but there is relatively little information on the contents of this "special shield". In the legend of Figure 6, the name "a specially constructed station for taking images during measurements" was used. It would be useful to have at least a schematic representation of a cross-section by "special shiel" (or "specially constructed station"?), to see to what extent the equipment fulfills the functions mentioned by the authors. Secondly, I think that the determination of the temperature values ​​was done by analyzing the thermographic images, so in a space outside the "special shield". Why then does the legend of Figure 6 mentions "taking images during measurements"?

Response:

Thank you for this remark. In order to illustrate the measurements with the special shield better, we have included an additional diagram in the article (Fig. 6b).

Remark 4

In Figure 8, distinct temperature values appear to be highlighted on the briquette. Can an arrow be inserted to highlight the direction along which the briquette is advancing, in order to correlate the temperature values with the initiation and development of the briquetting process, respectively?

Response:

Thank you for this remark. This was explained in the description of the Fig. 8.

Remark 5

Figures were misspelled, the first Figure 2 being, in fact, Figure 1.

Response:

This has been corrected in the article.

Remark 6

In lines 319-323, the authors state that “the temperature distribution on the surface of pillow-shaped briquettes correlates with research conducted by other scientists. Hryniewicz proved that the greatest unit pressure is exerted on the pillow-shaped briquette in its upper part [57], while Loginow et al. showed on an experimental stand that the highest deformations of the material occur in the upper part of this type of briquette [52]”. In fact, the results obtained by the authors of the proposed article seem to be somewhat consistent only with the results obtained by Hryniewicz.

Response:

Thank you for the remark. However, it should be pointed out that in  the experiments conducted by Loginow et al. the plasticine was used, not fine-grained material. in our opinion if a fine-grained material was actually used, it could be assumed that due to the greater deformation of the material, the stresses would also be higher at these points.

Remark 7

In Chapter 3. Results and Discussion, the discussion is mainly concerned with revealing the observed aspects, and only to small extent explanations or hypotheses are offered regarding the observed aspects.

Response:

Thank you for the remark. We have included the hypothesis in the Results and Discussion.

Remark 8

The formulation “The popular areas which successfully uses thermography is widely understood construction industry.” is confusing (The popular areas are widely ...).

Other less well-expressed forms in English: " due to it being non-invasive", ”in which case they range from CNC machine tools [12], through mechanical spreaders [13], rotating machines [14], belt conveyors [15], internal combustion engines (catalytic reactors) [16,17], not to mention bearings (temperature measurement of moving rolling elements, and one must keep in mind that baskets in bearings may be measured only with contactless methods”, ”Methods of measuring steel ropes were described which included the relationship between the growth of temperature and force”, “Phenomena concerning thermography a rich source of information related to technological processes and to changes or irregularities which take place during them”, etc.

  1. Authors need to pay more attention to editing the article and expressing it in English.

Thus, in lines 60 and 6,2 there are two formulations that start, both, with “After that, ...”.

It can be written "extrusion process of poly(vinyl chloride) " instead of "extrusion process of poly (vinyl chloride)", "it is rarely used to study consolidation processes." instead of “is rarely used to study consolidation processes.”, “a special” instead of “a spcial”, in line 193, “in Table” instead of “in Tables” in line 295, etc.

Line 254 reads "was equal -49 ° C." I think the expression is wrong, the temperature value being + 49 ° C.

In lines 270, 325, after the abbreviation "i.e." a comma must also be placed.

In the case of some of the figures (Figs. 2, 10, 14) no dot was placed at the end of the figures' legends.

There is a convention that some punctuation marks should be placed immediately after a word, without a blank space appearing. In lines 121, 122, etc., a blank space was placed before the comma.

The way in which the titles of some articles are included in the list of bibliographic references is uneven (in some cases, the so-called important words were written with the first letter as a capital letter - for example, references nos. 2, 42, while in other cases, except for the first letter of the title, only lowercase letters were used. In the list of bibliographic references, abbreviations of journal titles were used only in a few cases.

Response:

All of those remarks have been corrected in the article.

Round 2

Reviewer 3 Report

The revised manuscript has significant improvements.